# Smart touchless human–machine interaction based on crystalline porous cages

Jinrong Wang[1], Weibin Lin [1], Zhuo Chen [2], Valeriia O. Nikolaeva[1], Lukman O. Alimi [1] & Niveen M. Khashab [1,2] ✉

The rise of touchless technology, driven by the recent pandemic, has transformed human-machine interaction (HMI). Projections indicate a substantial growth in the touchless technology market, nearly tripling from $13.6 billion in 2021 to an estimated $37.6 billion by 2026. In response to the pandemic-driven shift towards touchless technology, here we show an organic cage-based humidity sensor with remarkable humidity responsiveness, forming the basis for advanced touchless platforms in potential future HMI systems. This cage sensor boasts an ultrafast response/recovery time (1 s/3 s) and exceptional stability (over 800 cycles) across relative humidity (RH) changes from 11% to 95%. The crystal structure's 3D pore network and luxuriant water-absorbing functional groups both inside and outside of the cage contribute synergistically to superior humidity sensing. Demonstrating versatility, we showcase this cage in smart touchless control screens and touchless password managers, presenting cost-effective and easily processable applications of molecularly porous materials in touchless HMI.

The onset of the COVID-19 pandemic since 2019 has severely disrupted humans' daily lives around the world[1,2]. This outbreak served as a poignant reminder of the rapid and perilous spread of viruses and bacteria. As a result, touchless human–machine interaction (HMI) is emerging as a new trend that promises to revolutionize the way humans interact with the devices around them. The rise of touchless HMI marks a new era in seamless and hygienic interaction with technology, providing innovative solutions across various applications[3–6]. The innovative technology enables individuals to perform tasks without the need for physical input, thereby eliminating the necessity of touching public surfaces such as doorknobs, elevator/ATM buttons, or shared screens. Specifically, smart sensors, which are critical components of touchless HMI systems, have the ability to accurately convert diverse information into signals that are recognizable by the machine. The increasing prevalence of touchless sensors highlights their vital role in shaping user experiences, enhancing convenience, and addressing hygiene concerns across a range of technological applications. Infrared (IR) sensors, adept at detecting emitted infrared radiation, excel in proximity sensing and gesture recognition. Ultrasonic sensors, utilizing sound waves for distance measurement, find applications in automatic doors and robotics. Capacitive sensors, sensitive to capacitance changes, play a crucial role in touchless interfaces like switches and interactive displays. Radar sensors, using radio waves, are instrumental in detecting object presence and movement. Lidar sensors, employing laser light for precise distance measurement, contribute to autonomous vehicles and 3D mapping applications. Optical sensors, including cameras, capture visual data for gesture recognition, facial detection, and augmented reality experiences. Magnetic sensors identify changes in magnetic fields, serving purposes in door/window position sensing and security applications. Acoustic sensors, relying on sound waves, contribute to touchless interfaces, and environmental monitoring. Among all these sensors, the humidity sensor, which is safe, comfortable, and easy to manufacture, seems to be the most appealing option for touchless HMI technology[7–11]. Unlike traditional touchless sensors that rely on different physical principles, the advantages of humidity sensors in

[1]Smart Hybrid Materials Laboratory (SHMs), Physical Science and Engineering Division, King Abdullah University of Science and Technology (KAUST), Thuwal 23955-6900, Saudi Arabia. [2]Advanced Membranes and Porous Materials Center (AMPM), King Abdullah University of Science and Technology (KAUST), Thuwal 23955-6900, Saudi Arabia. ✉e-mail: niveen.khashab@kaust.edu.sa

touchless HMI stem from their ability to capture and respond to a unique and natural aspect of human interaction-skin moisture. Human skin naturally releases moisture, and humidity sensors can capture variations in this moisture. This allows for a more natural and intuitive form of touchless interaction, as it leverages a biological parameter. The surface of human fingers is enveloped by a significant amount of water molecules, which can serve as a source of information for the body to realize touchless control. By exploiting the water molecules present on human finger surfaces as an input source, touchless control systems can facilitate contactless operation of devices, while minimizing the possibility of transmitting germs or diseases through physical touch. Ongoing exploration and progress in this field could potentially lead to new and innovative ways of interacting with technology and machines.

Nowadays, humidity sensors are extensively employed in diverse fields, such as human health, industry, agriculture, and environmental monitoring, where accurate detection and control of humidity levels are crucial. The utilization of various nanomaterials, such as metal oxides, polymers, carbon-based materials, and their composites, has generated considerable interest in the development of humidity sensors owing to their exceptional sensitivity towards water molecules[12–17]. However, some drawbacks still exist in terms of long-term stability, complicated preparation process, and unsatisfactory sensing performance. Recently, several attempts have been made to tackle these issues by developing new materials and proposing specific sensing mechanisms. One of the most gradually applied materials for humidity sensing include covalent organic frameworks (COFs) and metal-organic frameworks (MOFs), which are relatively young classes of porous crystalline solids[18,19]. Despite of remarkable stability and high water capacity, their poor solution processability limits their applications in sensing. In turn, organic cages have gained significant attention recently due to their intrinsic tunable nanocavities and rich structural diversity[20–25]. Because of their excellent solution-processability, which is distinct from MOF and COF, they can self-assemble in solution into defect-free crystalline porous materials. The rapid progress of supramolecular chemistry has resulted in the discovery of a growing range of organic cages and their widespread applications, such as molecular recognition, catalysis, separation, and storage[26–31]. Several attractive qualities, such as chemical robustness, range of functionalities, intrinsically adjustable crystal porous structure, and appropriate proton conduction[32,33], make organic cages promising in sensing applications. However, there is currently no report on any system exploiting intrinsically porous organic cages that can effectively respond to changes in humidity. Thus, studying the performance of organic cages-based sensors alongside their crystalline and structural features could open up a brand-new pathway for humidity sensing.

Herein, we present a different approach to humidity sensing that utilizes a unique organic cage featuring carboxylic and protonated amine functional groups (Fig. 1a)[34]. The cage has a 3D crystal structure and luxuriant water-absorbing functional groups, resulting in exceptional humidity sensing capabilities. An ultrafast response/recovery time (1 s /3 s) and remarkable stability (consistent operation over 800 cycles) were achieved with an extreme response on varying RH from 11% to 95%. The excellent humidity responsibilities are reached due to the self-assembly of the organic cage with highly interconnected network and widespread H-bonding anchors in both the inner and the outer arrangement of its cavity. Additionally, we compared the performance of the 3D carboxylic cage (Cage-1) with a cage lacking carboxyl groups (Cage-2) and a common 2D trianglamine macrocycle (TA) in terms of their humidity stimuli-responsiveness. Our study revealed significant differences in the humidity sensing properties of the selected samples. Results from the single crystal X-ray diffraction (SCXRD) indicated that the −COOH modified protonated Cage-1 with

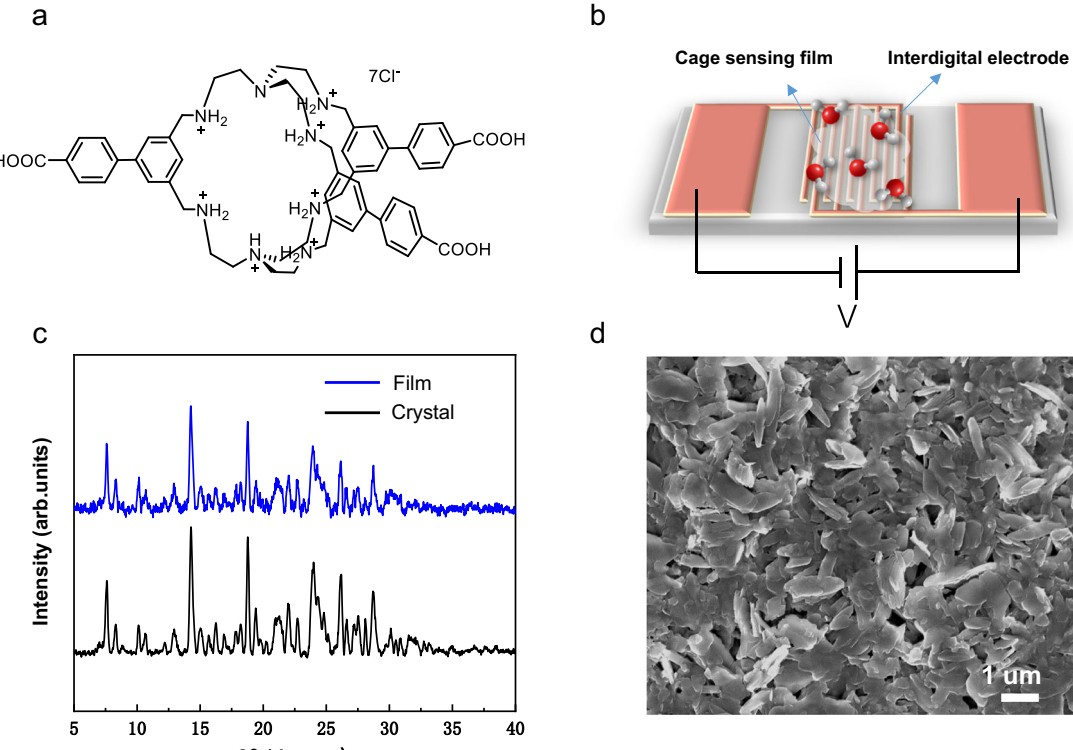

**Fig. 1 | Chemical structure of the cage and crystallinity of the cage sensing film.** **a** Cage-1 chemical structure. **b** A schematic depiction of a Cage-1-based humidity sensor. The RH-sensitive device resistance R (RH) is recorded upon application of a constant DC. **c** PXRD of the casted Cage-1 sensing film and cage crystal. **d** SEM image of the Cage-1 sensing film deposited on the interdigital electrode.

more H-bonding anchors self-assembled into a supramolecular conductor with 3D water channels for efficient water transfer, which made this cage-based system more susceptible to water molecules when exposed to variable humidity conditions. Furthermore, simulations confirmed that this system with its 3D interconnected network is ideal for fast water diffusion and transport. In addition, a state-of-the-art touchless control screen and password manager have been successfully developed using a specially designed cage-based 5 × 5 sensor array that showed an exceptional operation and reliability in real-life applications. This represents a significant step forward for touchless HMI systems and opens up promising opportunities for molecularly porous materials as easily processable recognition units.

## Results

The precise chemical structure of the carboxylic and protonated amine cage complex is shown in Fig. 1a and confirmed by X-ray crystallographic analysis[34]. Details on the cage synthesis and characterization are shown in Method part, Supplementary Fig. 1 and Supplementary Fig. 3, 4. The organic cage is designed to contain a high number of moisture-sensitive functional groups, namely amine groups within the cavity and carboxylic groups on the external part of the unit. These sensitive moieties create extensive H-bonding anchors throughout the interior and exterior of the cage, producing a dense network of proton-hopping pathways. When these moisture-sensitive groups react with water molecules, they alter the proton conductivity of the cage material, resulting in corresponding changes in the electrical signal. Figure 1b illustrates a schematic diagram detailing the components of the humidity sensor. Unlike many other COF and MOF materials, the cage material in our study exhibits good water solubility (Supplementary Fig. 10). This unique characteristic significantly simplifies the preparation of cage sensors, making the process notably straightforward. The as-synthesized cage molecules (10 mg/mL in water) aqueous solution were drop casted into a sensing film on the surface of an interdigital electrode. The details of the device fabrication process and sensing test technology are shown in Supplementary Figs. 10, 11. The high crystallinity of the cage sensing film was well maintained, as evidenced by the PXRD results (Fig. 1c). The morphology of the cage crystalline thin sensing films is represented by submicron lamellar structures (Fig. 1d) markedly different from the as-prepared cage powder (Supplementary Fig. 12). The observed topological distinction can be traced back to the film preparation process, where the cage aqueous solution was deposited onto the electrode surface via drop-casting and then subjected to a heating treatment. Notably, the recrystallization process that occurs during water evaporation plays a critical role in the formation of lamellar crystalline sensing film on a submicron level. This phenomenon is of utmost importance as it contributes to the development of a continuous and homogeneous film structure, which ultimately ensures topological consistency across the entire device surface, leading to improved humidity sensing stability. We employed a scanning electron microscope (SEM) to measure the thickness of the sensing film, yielding a thickness of approximately 6.2 μm. The SEM images distinctly illustrate the exceptional uniformity of the sensing film (Supplementary Fig. 13).

Electrochemical sensors are known to display remarkable changes in electrical properties with a slight shift in the surrounding environmental conditions. Humidity sensors are generally categorized into resistive, impedance, capacitive and voltage types depending on various measurement strategies. This study is focused on a resistive sensor, which operates easily and consistently in direct-current (DC) mode. Various saturated salt solutions could generate a well-controlled humid environment with certain RH levels. In this regard, 95%, 85%, 75%, 67%, 43%, 33%, 23%, and 11% RH conditions were produced based on saturated $KNO_3$, KCl, NaCl, $CuCl_2$, $K_2CO_3$, $MgCl_2$, $CH_3COOK$, and LiCl aqueous solutions, respectively (Supplementary Fig. 11).

To evaluate the sensing performance of the cages under different humidity conditions, the device was exposed dynamically and continuously for 20 s at different humidity levels. As depicted in Fig. 2a, an increase in humidity level results in a significant decrease in sensor resistance, particularly at low RH levels. This feature is advantageous for humidity sensors that are intended for ultra-dry applications. The variations in sensor resistance at different humidity levels are intricately linked to the proton conductivity of the sensing material. According to the Arrhenius equation, proton conductivity in the cage sensor is influenced by both the proton conduction mechanism and the concentration of proton carriers[35]. In conditions of low humidity, the cage surface hosts only a minimal amount of water molecules, primarily adsorbed through hydrogen bonding with protonated amine and carboxyl groups. Hindered by these hydrogen bonds, water molecules experience restricted mobility, and the sparse arrangement of water molecules prevents the formation of a continuous network of hydrogen bonds. Proton transfer demands substantial energy, and the low number of charge carriers contributes to high resistance in the cage sensor. With an increase in humidity, more water molecules accumulate in the cage material. Under the influence of the electrostatic field, water molecules undergo ionization, yielding a substantial number of charge carriers. Facilitated by the Grotthuss chain reaction ($H_2O + H_3O^+ = H_3O^+ + H_2O$)[36], proton transfer occurs readily within the cage. This amplifies proton conductivity, leading to a reduction in the resistance of the cage sensor. The dynamic interplay between humidity levels and the proton conduction mechanism elucidates the responsiveness of the cage sensor to varying environmental conditions. Proton conductivity measurements were conducted at 303 K using compacted pellets of the cage. The results indicate that the proton conductivity exhibited an increase with RH within the range of 67–95% (Fig. 2b). The maximum value recorded was $1.28 \times 10^{-3}$ S cm$^{-1}$ at 95% RH (Supplementary Tab 1). The response of the cage sensor to the humid environment of saturated salt solutions and recovery towards indoor RH = 60% were also examined (Supplementary Fig. 14). During the tests, the indoor humidity level was measured by a commercial humidity sensor, which remained around 60%. The results of the experiment show that the resistance values of the cage humidity sensor increase when the detected humidity level is lower than the indoor humidity level of 60%, as shown in Supplementary Fig. 14a–d. On the other hand, the resistance values decrease when the detected humidity level is higher than 60%, as shown in Supplementary Fig. 14e–h. The variations in the humidity levels also affect the response speed and response value of the sensor. Overall, the cage humidity sensors can be advantageous in monitoring ambient humidity levels in various applications.

To demonstrate the diverse responsiveness of humidity sensors in different humidity conditions, we developed a water surface humidity detector based on the Cage-1 humidity sensor (Fig. 2c). The humidity level gradually decreases as it moves upwards from the water surface due to water evaporation. The influence of the distance between the water surface and the sensor on the resistance value change was examined based on this phenomenon. The response of the cage sensor to the humid vapor with the different distance levels (h1, h2, h3) and recovery towards indoor RH = 60% were examined over 5 cycles. Shorter distances provide higher relative humidity values, resulting in a decrease in the resistance change value as the distance increases from h1 to h3. Based on this design and the mechanism of humidity affecting resistance signal changes, the cage sensor has great potential for practical applications in real-time accurate detection of humidity in the environment.

As was previously assumed, the protonated carboxylic amine Cage-1 complex (Fig. 3a) has several advantages that make it attractive for humidity sensing. These include the formation of H-bonding networks with 3D water molecule channels and abundant H-bonding active sites inside and outside of the cage, allowing a faster response to

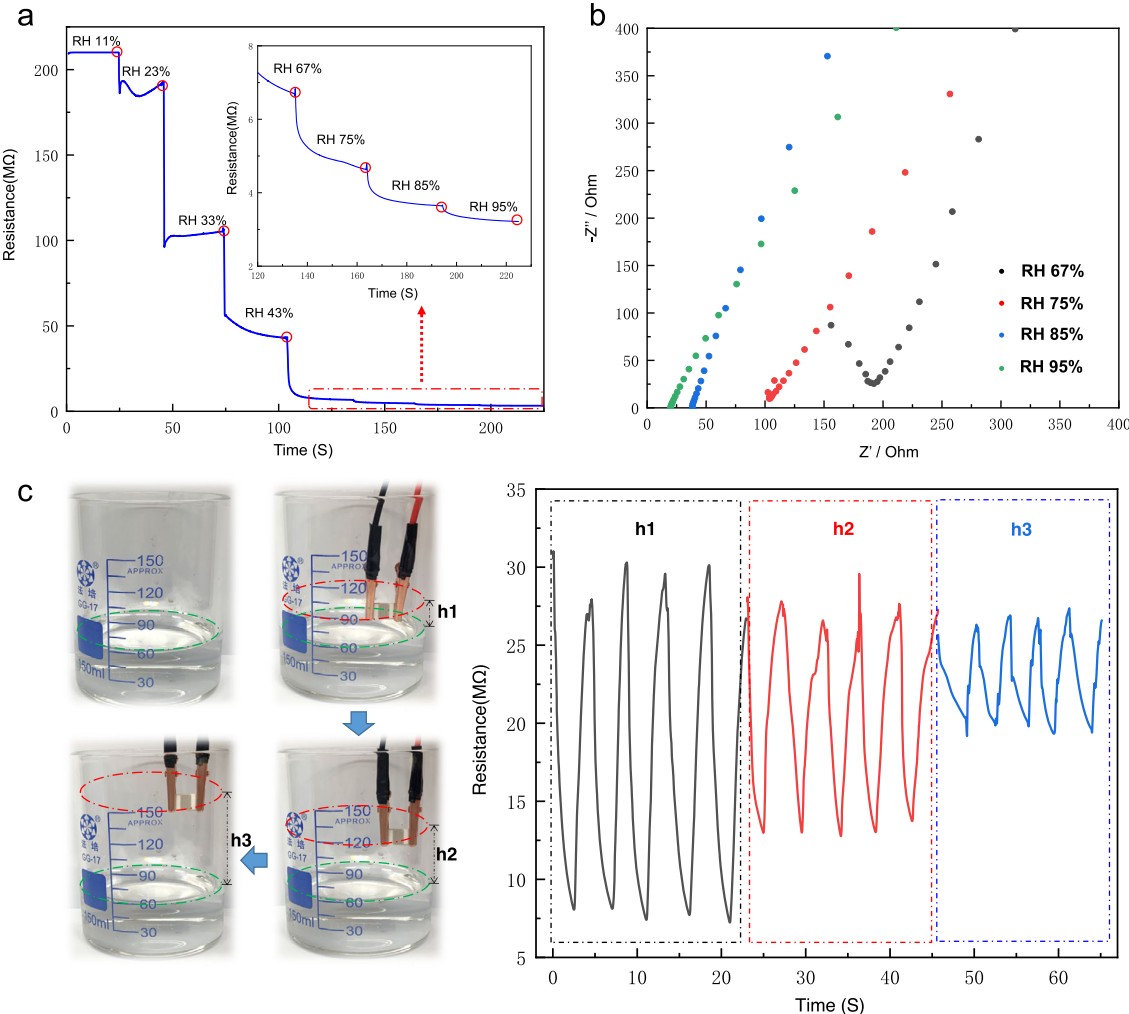

**Fig. 2 | Humidity sensing properties of the cage sensor.** Dynamic response curves of (**a**) the resistance change over time under the RH levels range from 11% to 95%, (**b**) Nyquist plots showing the impedance of Cage-1 at 303 K with varying RH, (**c**) photographs of the Cage-1-based sensor being placed near the water surface and removed afterwards, corresponding resistance curves measured over time.

water molecules. To demonstrate the role of carboxyl groups outside of the cage in humidity sensing, an identical cage without carboxyl groups (Cage-2) was designed and synthesized (details in Supplementary Fig. 2 and Supplementary Fig. 5, 6) for contrast, as shown in Fig. 3b. Furthermore, to demonstrate the contribution of 3D networks, the humidity-responsive performance of the Cage-1 was compared to that of a common 2D trianglamine (TA) macrocycle, as shown in Fig. 3c. The synthesis of TA was carried out as previously reported[37]. Details on the TA characterization are shown in Supplementary Fig. 7, 8. The PXRD results clearly demonstrate the good crystallinity of the Cage-2 and TA sensing films (Supplementary Fig. 9). Overall, these comparisons help to illustrate the unique advantages of the protonated carboxylic amine cage complex for humidity sensing.

One of the most crucial parameters of a humidity sensor is the sensing speed to satisfy the environmental conditions in practice. Significant differences were detected between the above three samples in terms of response and recovery speed. These materials were statistically compared in terms of their response and recovery speeds during the first humidity response cycle (as shown in Supplementary Fig. 15). The measured response time of the Cage-1 sensor was 1 s when RH changed from 11% to 95%, while the recovery time was 3 s when RH reduced back from 95% to 11%, which was about 2.6 and 11.4 times faster than those of the Cage-2 sensor, respectively (Fig. 3d, e), indicating that the carboxyl groups located on the outside of this organic

cage play an important role in humidity sensing. A detailed explanation is provided in the mechanism section. The measured response time for the TA sensor was 7.8 s, while the recovery time was as long as 30.1 s (Fig. 3f). These demonstrate 3D structure of the cage does help to accelerate the adsorption and desorption of water molecules. Reproducibility and cycle stability are equally essential parameters for evaluating the performance of humidity sensors. To measure the cycle stability of the manufactured devices, the sensor response was recorded under high humidity levels (95% RH) and its recovery under low humidity levels (11% RH). The Cage-1 sensor exhibited excellent cycle stability without performance degradation (Fig. 3g) even after more than 800 cycles (Supplementary Fig. 16), supporting its robust nature. By contrast, the humidity-sensing signals of the Cage-2 sensor and the TA sensor decayed substantially during cycling, indicating poor cycling stability (Fig. 3h, i).

Additionally, the durability of the cage sensor was tested by exposing it to an indoor environment with 60% RH for 3 months. As shown in Supplementary Fig. 17, the Cage-1 sensor maintained its excellent sensing performance with RH levels changing between 11% and 95%, demonstrating the high stability of the sensor. It is important to note that long-term exposure of humidity sensing material to high levels of humidity can cause irreversible damage to its structure and sensing capabilities. To evaluate the sensor's resistance to high humidity levels, it was tested after exposing it to 95% RH for 36 h, as

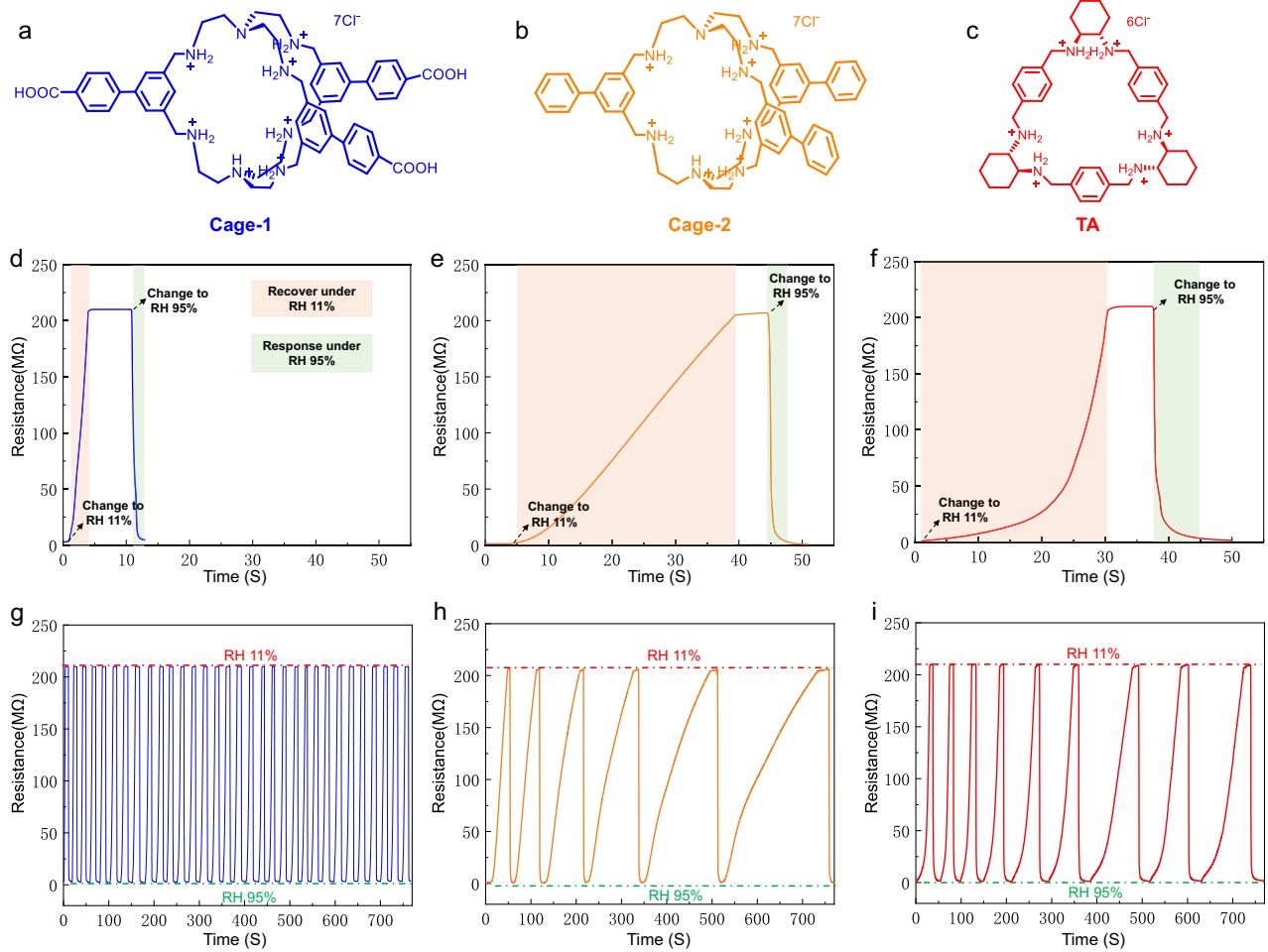

**Fig. 3 | Chemical structure and humidity sensing performance comparison.** The chemical structure of (**a**) Cage−1, (**b**) Cage-2, and (**c**) TA. The humidity sensing properties of (**d**) Cage-1, (**e**) Cage-2 and (**f**) TA with the RH changing from 11% to 95%, respectively. Reproducibility and cycle stability of the (**g**) Cage-1, (**h**) Cage-2, and (**i**) TA over various cycles of humidification (RH 95%) and desiccation (RH 11%).

shown in Supplementary Fig. 18, and no discernible performance degradation was observed. Furthermore, we performed PXRD analysis on the cage both before and after the humidity sensing experiments. The results indicate no significant change in the structure of the Cage-1 before and after the humidity sensing experiment (Supplementary Fig. 19). This further validates the structural stability of the Cage-1, which could play as a stable framework and platform for humidity sensing. Supplementary Table 2 compares various humidity sensors in terms of response/recovery speed and cycle stability. It has been observed that our Cage-1 humidity sensor demonstrates exceptional sensing speed and remarkable cyclic stability, essentially meeting the criteria for touchless HMI applications.

To obtain a deep insight into the molecular mechanism in detail, crystallographic analysis and simulations were adopted and performed. Diffraction grade crystal of the protonated Cage-2 was obtained via slow evaporation of water/methanol/dichloromethane mixtures at room temperature for comparison. Together with crystals of Cage-1 and TA[34,37], the molecular packing architecture and interconnected voids were explored at the molecular level (Supplementary Fig. 20, 21)[38,39]. Although Cage-1 and Cage-2 both exhibit six secondary amines, one tertiary amine, and 7 counter chloride anions, the introduction of three carboxylic acid groups endow Cage-1 with 3D structure more potential hydrogen bond sites and has a great influence on the crystal packing and supramolecular interactions. As shown in Fig. 4a, these −COOH groups interact with chloride anions to expand the void space in the crystal, which thereafter brings about the 3D

interconnected water channels. In comparison, the Cage-2 without −COOH groups shows a compact crystal packing style with only 2D layered water network (Fig. 4b, Supplementary Fig. 20). As for the 2D macrocycle, just 1D water channels are formed in its crystal packing mode (Fig. 4c). The distinct water channels would produce different porosity network for these materials after activation (Fig. 4d–f). Thus, continuous and isolated voids were further analyzed for these materials after activation, which could provide a deeper insight about their different porosity.

As shown in Fig. 4g–i and Supplementary Fig. 24, the fraction of voids with different sizes that are interconnected/isolated is analyzed and compared quantificationally. For this simulation, probes with different radii (0.60, 0.80, 1.0, and 1.2 Å) were inserted. When the small probes with a radius of 0.60 Å or 0.80 Å are inserted (Supplementary Fig. 24), the voids of all three materials are highly interconnected (green). Nevertheless, compared with the considerable disconnected part (red) in Cage-2 and TA, the Cage-1 film has the highest porosity for easily diffusion of small molecules. As the probe radius increases, the fraction of interconnected voids decreases. When a theoretical probe radius of 1.0 Å is used, disconnected voids (red) fully dominate in Cage-2 and TA (Fig. 4h–i and Supplementary Fig. 24), while Cage-1 still shows good interconnected porosity (green, Fig. 4g). High microporosity and interconnectivity are expected to cause continuous water vapor flow and influence the adsorption/desorption process. This simulation could be interpreted as being exclusively related to the diffusivity of Cage-1 material and how it contributes to enhancing the transport of

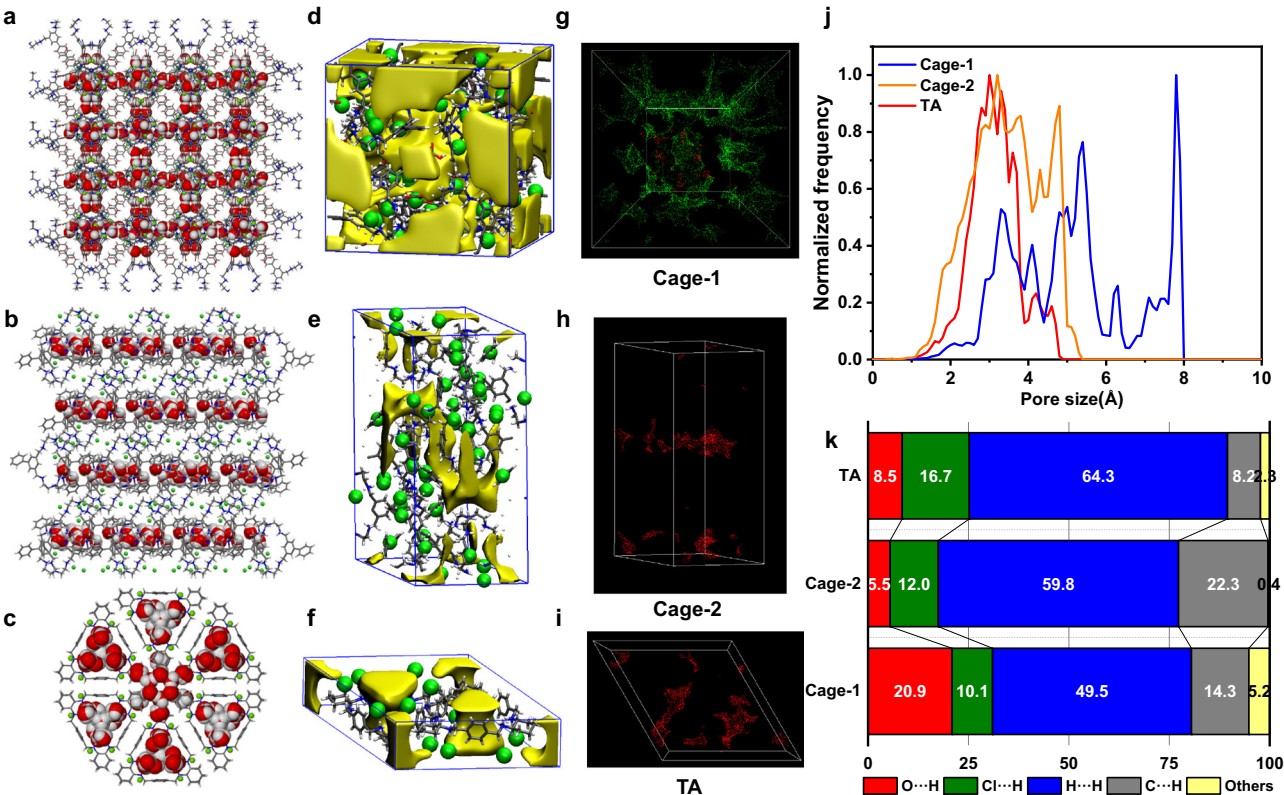

**Fig. 4 | Comparison of pore structure and supramolecular interactions analysis in Cage-1, Cage-2 and TA.** Crystal packing showing the water channels and networks of (**a**) Cage-1, (**b**) Cage-2 and (**c**) TA. Element color code, Green, Cl; red, O; blue, N; White, H; gray, C. Simulated interconnected free volume showing intrinsic cavity and extrinsic voids of (**d**) Cage-1, (**e**) Cage-2 and (**f**) TA crystal after activation. Interconnected (green) and disconnected (red) voids in (**g**) Cage-1, (**h**) Cage-2 and (**i**) TA with respect to a probe of 1.0 Å radius. **j** Simulated pore size distributions of Cage-1, Cage-2 and TA. **k** Relative contributions of various interactions to the Hirshfeld surface area of hosts Cage-1, Cage-2, and TA in their crystals.

water molecules over those of Cage-2 and TA materials. Furthermore, the exact voids size distributions of TA, Cage-1 and Cage-2 can be easily derived from the crystal structures (Fig. 4j). The most frequently present voids of TA show an average size of 3.0 Å, which may result from the intrinsic cavity of TA with chloride. In contrast, Cage-2 shows another pore size peak around 4.8 Å besides 3.3 Å formed by the intrinsic cavity, which probably caused by the extrinsic voids. While situation in Cage-1 is more colorful, showing a widespread pore size distribution from 3.3 Å to 7.8 Å. These multiple voids present not only the intrinsic cavity, but more importantly the various extrinsic voids formed by this special Cage-1 chemical structure. Together with the highly interconnected microporosity, the widespread and larger void sizes endow Cage-1 better ability for water molecules permeance. The simulation does not consider the possible defects, swelling of the film, and fragment shift after activation, which could potentially alter the absolute sizes of cavities and pores. However, the simulation reveals the high free volume and porosity of the Cage-1 material to an extent, proving the significance of the cage building blocks with −COOH groups in creating permanently interconnected nanoporosity within the crystalline solid material for the diffusivity of permeants and quick water adsorption/desorption.

The analysis of the calculated Hirshfeld surfaces and the subsequent fingerprint plots could provide together the qualitative and quantitative presentation of the contribution of the supramolecular interactions within the host-guest crystal system[40]. As shown in Fig. 4k, a direct presentation was depicted showing the contribution percentages of various host-guest interactions. In particular, hydrogen bonding O···H play an important role in Cage-1 (20.9%), which is more than twice of TA (8.5%) and nearly four-fold of Cage-2 (5.5%). This result indicates much more water affinity to Cage-1, showing stronger

supramolecular interactions between water molecules and Cage-1 host. Besides, the Cl···H percentage contributions of Cage-1 is just 10.1%, smaller than those of Cage-2 (12.0%) and TA (16.7%). Combined with the results from their Hirshfeld surface mapped with $d_{norm}$ (Supplementary Fig. 22, 23), it is found that most chloride anions interact with −NH groups, which are good binding sites for water molecules, especially in Cage-2 and TA. However, the negative effect of chloride anions was significantly restrained by −COOH groups in Cage-1 material. As shown in Supplementary Fig. 23, the three −COOH groups could bind strongly with three chloride anions, which releases more −NH sites for stronger hydrogen bonding O···H to realize better water molecule interactions. Hence, the introduction of −COOH groups not only help Cage-1 to construct the 3D highly interconnected pore networks with good diffusivity based on both the intrinsic cavity and extrinsic void, but also build up its supramolecular interactions between the cage host with water molecules. The overall effect undoubtedly endows Cage-1 with efficient ability for water transfer and sensing.

In order to compare the plausible mechanisms of proton conduction in the three molecules studied, we measured the proton conductivity of Cage-1, Cage-2 and TA by using alternating current electrochemical impedance spectroscopy (EIS) at different temperatures (303−343 K) under a fixing RH of 85% (Fig. 5a−c, Supplementary Tab 4). As is known, proton conduction involves two main mechanisms, which are the Grotthuss and vehicular mechanisms. Typically, a reference activation energy value of 0.4 eV is used, where higher or lower values correspond to the vehicular and Grotthuss mechanisms, respectively[41,42]. To determine the activation energy (Ea) for proton conduction, we performed a least-squares fit of the Arrhenius plots using the temperature-dependent conductivity at 85% RH, as

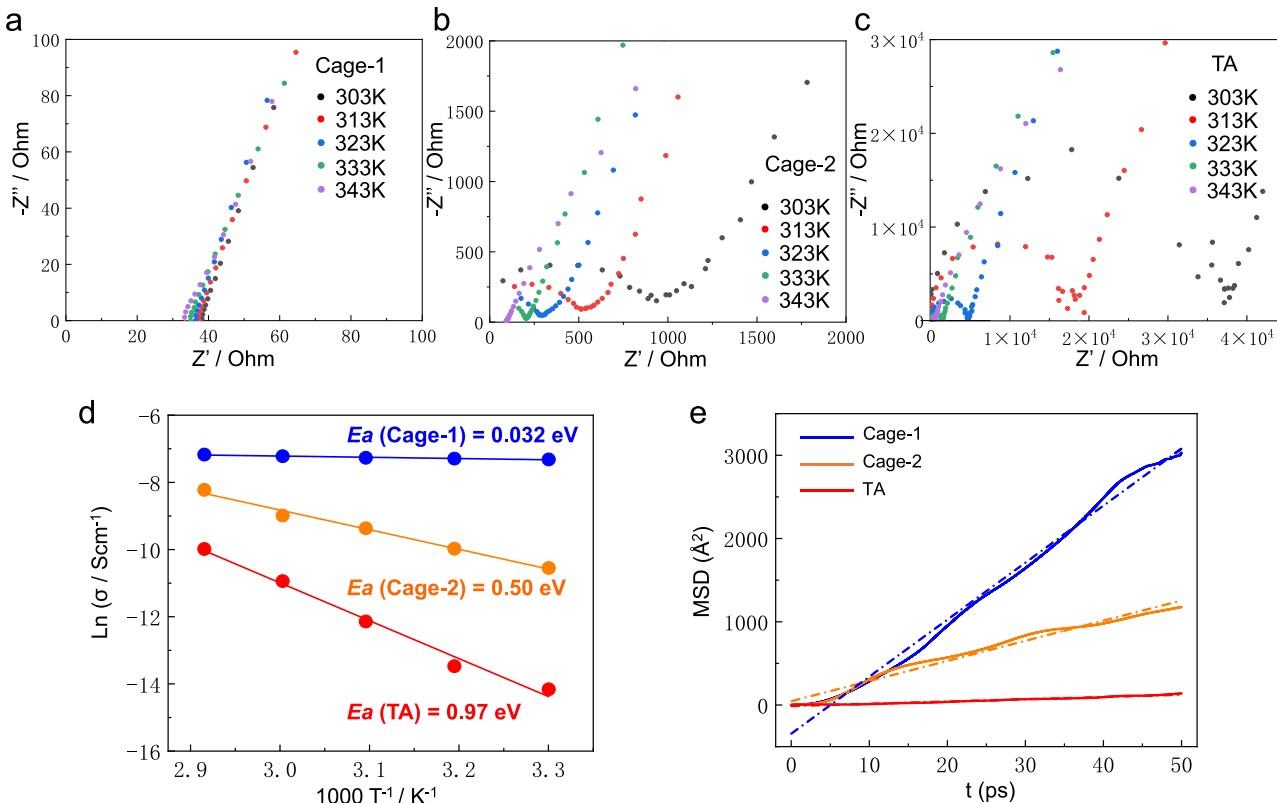

**Fig. 5 | Comparison of electrochemical data, proton conductivity and average mean square-displacements (MSDs) for Cage-1, Cage-2 and TA.** Nyquist plots of (**a**) Cage-1, (**b**) Cage-2 and (**c**) TA as a function of temperature (303 – 343 K) under fixed humidity 85% RH. **d** Arrhenius plots of the proton conductivity of Cage-1 (blue), Cage-2 (yellow) and TA (red) under 85% RH. Least-squares fittings are shown as blue, yellow and green solid lines, respectively. **e** Comparison of the simulated MSDs of Cage-1, Cage-2, and TA.

illustrated in Fig. 5d. The remarkably low activation energy of 0.032 eV for Cage-1, derived from the Arrhenius plot, undoubtedly suggests a Grotthuss mechanism. In this mechanism, a hydronium ion reorients and passes its proton to a neighboring water molecule through hydrogen bonds. The interconnected 3D pore architecture and abundance of hydrogen bonding sites ($-NH_2^+$ and $-COOH$) in Cage-1 play a crucial role in facilitating fast proton transport. In contrast, Cage-2 and TA exhibit vehicular mechanisms, as indicated by their higher activation energies of 0.50 eV and 0.97 eV, respectively. In vehicular mechanisms, only water molecules serve as carriers for proton transfer along the pore channel. Additionally, the proton transport pathways in Cage-2 (2D network) and TA (1D channels) are not as effective as the 3D porous architecture in Cage-1 for water diffusion to realize efficient proton transfer.

To further understand the mechanism of water dynamics and water diffusion in the crystalline phases of Cage-1, Cage-2 and TA, ab initio molecular dynamics (AIMD) simulations were adopted and performed at a temperature of 500 K for up to 50 ps by CP2K package (see the Simulations part in Method and Supplementary Fig. 25)[43]. Due to the large sizes of the organic macrocycle or cage molecules, we focused on the diffusion of water molecules. As shown in Fig. 5e, it is found that average mean square-displacements (MSDs) of Cage-1 is much larger than those of Cage-2, and especially TA macrocycle. Specifically, the diffusion coefficients of water molecules ($D_{water}$) in Cage-1 ($D_{water} = 1.14 \times 10^{-3}$ cm$^2$ s$^{-1}$) is nearly threefold of Cage-2 ($D_{water} = 4.05 \times 10^{-4}$ cm$^2$ s$^{-1}$), and two orders of magnitude higher than that of macrocycle TA ($D_{water} = 4.52 \times 10^{-3}$ cm$^2$ s$^{-1}$, Supplementary Fig. 25). The faster diffusion of water endow the Cage-1 system with quicker and better water recognition/transport ability, which is in good accordance with its Grotthuss mechanism. Therefore, the crystal structure formed in Cage-1 with 3D highly interconnected pore

networks is a promising and highly efficient platform for water diffusion and water sensing.

Collectively, this Cage-1 humidity sensor exhibits tremendous potential in touchless sensing applications, particularly in detecting human respiratory behavior. To showcase this potential, a touchless human respiration monitor was developed using the Cage-1 humidity sensor, as depicted in Supplementary Fig. 26. The humidity sensor was placed near the experimenter's nose, where it was able to detect water vapor molecules in exhaled air, leading to changes in resistance signals. Three different respiratory behavior types were examined, each yielding significantly different sensing signals. Deep breathing, for instance, resulted in a larger and relatively slower variation in resistance signal values due to the increased humidity state and longer recovery time provided to the cage humidity sensor. Conversely, fast breathing provided faster response and recovery speed, resulting in a faster and relatively smaller change in resistance signals. Given the cage humidity sensor's ability to detect notably different resistance signal changes for various respiratory behaviors, it can be used for real-time monitoring of respiratory-related illnesses such as congenital asthma in infants and respiratory distress.

As previously discussed, touchless human–machine interface (HMI) has emerged as an intelligent approach for novel communication between humans and machines, overcoming the limitations posed by contact control. This is particularly relevant since human finger surfaces are enveloped by a large number of water molecules, thus making it convenient and suitable for touchless HMI control. To evaluate the touchless sensing capabilities of the Cage-1 humidity sensor, Real-time R-t curves displayed that the resistance of the cage humidity sensor reduced when the finger approached to the sensor surface, and increased rapidly when the finger moved away, highlighting its impressive humidity sensitivity, fast response and recovery speed

(Supplementary Movie 1). The tests revealed consistent responses and recoveries, confirming the high stability, fast response, and recovery speed of the touchless fingertip sensor when the finger approached and withdrew from the sensor surface. We systematically moved the finger away from the humidity sensor, conducting touchless sensing at various distances (2 mm, 4 mm, 6 mm, and 10 mm) between the finger and sensor surface for 9 consecutive cycles respectively (Fig. 6a). The results demonstrate that the resistance response of the Cage-1 sensor gradually decreases as the distance between the human finger and the sensor increases. Importantly, the response and recovery times remain stable throughout, confirming the humidity sensor's effectiveness in touchless sensing applications. The RH value around the surface of the human finger is greater than the indoor RH value, which is a prerequisite for the normal operation of our designed cage-based finger humidity sensor. Therefore, the sensing ability of the fingertip humidity sensor becomes more prominent in testing environments with lower relative humidity. Additionally, we observed no changes in resistance when a gloved finger or a hot metal surface with a temperature of 70 °C approached the sensor's surface, as shown in Supplementary Fig. 27. These results confirmed that the signal change was caused by water molecules surrounding the finger and eliminated any influence of finger temperature on the finger humidity sensor. Furthermore, to assess the robust durability of the sensor, an abrasion test was initiated, subjecting the sensing film to vigorous rubbing by a finger dozens of times. Subsequently, no detachment of the sensing film was observed as shown Supplementary Movie 2. The sensing film exhibited commendable interfacial adhesion to the sensor substrate after drop-casting and drying, confirming its robust durability. Subsequent finger humidity sensing tests were also performed and the sensing signal was not attenuated by contact friction (Supplementary Fig. 28), which proved the excellent stability of the Cage-1 sensor.

Figure 6b, c illustrates the design flowchart and schematic diagram of the HMI system for our touchless control screen. The system incorporates a smart sensor array matrix, wireless modules, and an electronic screen (Supplementary Fig. 29). The printed circuit board (PCB) layout design of the touchless control screen device is presented in Fig. 6d, while the circuit diagram for the main modules is provided in Supplementary Fig. 30. By virtue of the excellent sensitivity of our cage fingertip humidity sensor, the cage humidity sensing array can track the touchless movement trajectory of the human fingertip on its surface. When the finger humidity is detected by the cage sensor array, humidity information is transformed into the analog signal through the drive module in the way of resistance voltage division. The analog signal passes through the voltage follower and is converted into a digital signal by the Analog-to-Digital Converter (ADC). The digital signal is further processed and calculated by the Microcontroller Unit (MCU) and transmitted through the wireless module. Finally, the signal is uploaded to the screen through the signal conversion module, achieving a real-time display of fingertip movement trajectories on the electronic screen. The touchless control screen can realize three modes of detection, which are continuous glide gesture monitoring (Fig. 6 e and Supplementary Movie 3), intermittent tap monitoring (Fig. 6e and Supplementary Movie 4), and line click monitoring (Fig. 6e and Supplementary Movie 5). To effectively discriminate the complex gliding motion of the fingertip, the touchless electronic screen was applied as a writing tablet. The numbers "1,2"and the Chinese character "王" were successfully written by controlling the movement trajectory of the human finger (Fig. 6e and Supplementary Movie 6–8). These results highlight the significant potential of our smart cage humidity sensing array as a touchless controlled screen for remote interaction applications. As another demonstration of the smart cage sensor array's capabilities, we applied it to a touchless password manager. For this purpose, we set a specific movement trajectory as the password beforehand. As depicted in Fig. 6f, if the finger movement trajectory is consistent with the password, the screen will display a happy face indicating that the password matches, otherwise, an angry face is displayed to indicate a wrong password (Supplementary Movie 9, 10). This touchless control password manager offers a smart interaction method that can protect privacy by avoiding contact with fingerprints. Furthermore, we conducted cyclic stability and time stability tests on the device. Following a hundred of demonstrations, the device showcased good repeatability. Notably, even after an extended period of six months, the device maintained remarkable repeatability.

It's crucial to note that our current device represents an initial iteration, leaving ample room for refinement and enhancement in subsequent developments. To enhance the sensitivity of the device, advancements can be pursued on two fronts. Firstly, in the realm of materials, there is potential for further elevating the response speed of the sensing material itself. Secondly, on the hardware side, it can be achieved by improving the channel switching speed of analog switches, increasing the sampling rate of ADC, and upgrading to a touch screen with a sufficiently fast response speed.

## Discussion

In this work, we report an organic cage-based humidity sensor with exceptional performance capabilities. By taking advantage of the abundant functional sites for hydrogen bonding and the interconnected 3D water channels within the organic cage crystal, an ultrafast response/recovery speed and excellent sensing stability were achieved. Results from the single crystal and simulations indicated that the protonated Cage-1, with the introduction of –COOH groups, self-assembled into a supramolecular H-bonding network with highly interconnected porosity, which made this cage-based system more efficient for water transfer and susceptible to the water molecules when exposed to variable humidity conditions. Besides, a human fingertip humidity sensor was further fabricated with great performance, showing the excellent capabilities of the cage-based sensor for effective and reliable touchless sensing. Finally, the touchless fingertip humidity sensor was successfully applied as a touchless screen and touchless password manager, opening a promising future for employing molecularly porous materials in touchless HMI applications.

## Methods
### Materials
All commercial reagents were obtained from Sigma-Aldrich and used as received without further purification. Aqueous solutions were prepared with Milli-Q water. Flash column chromatography was performed on 200-300 mesh silica gel. Thin-layer chromatography (TLC) was performed on precoated silica gel aluminum plates and observed under UV light.

### Instrument and characterization
$^1H$ and $^{13}C$ spectra were recorded on Bruker AVIII 400 MHz or 500 MHz spectrometers at room temperature. Powder X-ray diffraction (PXRD) analyses were performed using a Bruker D8 Advance X-ray diffractometer at room temperature. The morphology of the cage powder and sensing film were explored by scanning electron microscopy (SEM) (Magellan SEM, FEI). Commercial humidity (HIH-4000-003 from Honeywell) sensors were used as references. The humidity sensing measurements were carried out using a Keithley 2400 SourceMeter unit. Impedance spectroscopy was employed for proton conduction measurements by using an EC Labs Biologic VMP3 potentiostat. A flexible interdigitated electrode array is detailed with specific characteristics. The substrate material is Polyimide (PI) and has a thickness of 0.25 mm. The overall dimensions of the array measure 50 mm by 74 mm, featuring a total of 5 × 5 units. Each unit is defined by a finger width of 100 μm, a spacing of 100 μm, and a finger length of 3 mm, with a total of 11 pairs of fingers. The metal layer structure comprises Cu and Au with thicknesses of 12 μm and 25 nm, respectively. This array

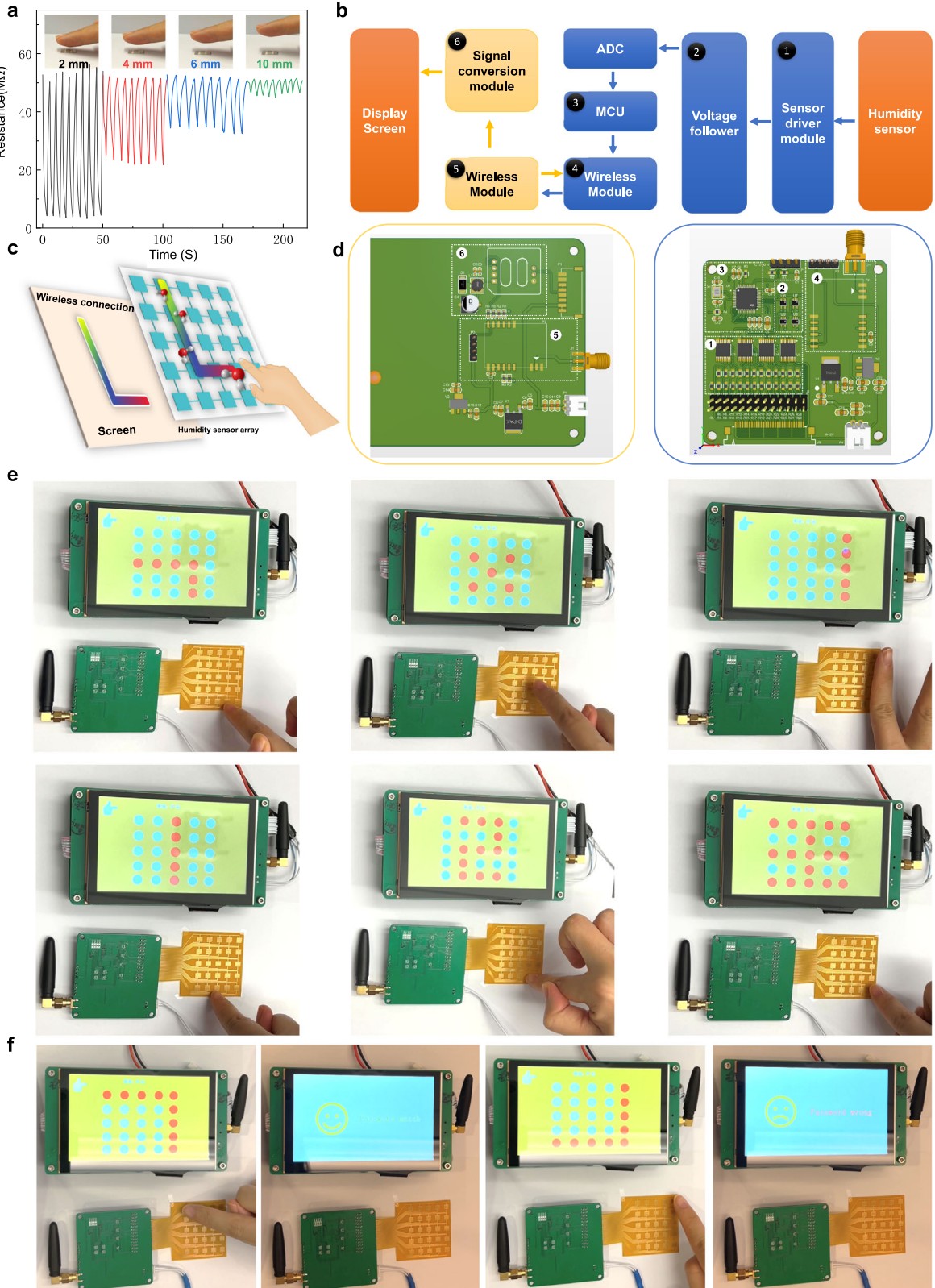

**Fig. 6 | Demonstration of the smart touchless HMI application based on cage humidity sensor. a** Relationship between the humidity sensing and the finger-to-sensor distance. **b** Design flowchart of the HMI system for touchless control screen. **c** Schematic diagram of the smart cage sensor arrays for touchless control screen. **d** PCB layout design of the touchless control screen. **e** Demonstration of the touchless control screen. **f** Demonstration of the touchless password manager. For all demonstrations, the distance between the finger and the sensor is controlled at 4–6 mm.

is designed to provide flexibility and functionality for touchless HMI applications. 5-inch capacitive touch intelligent serial screen with IPS wide-angle screen and resolution of 800 × 480. 433 MHz wireless serial transmission and reception RF module.

## Synthesis of protonated Cage-1

To a solution of 5-bromoisophthalaldehyde (1.608 g, 6 mmol) in 100 mL acetonitrile was dropwise added with tris(2-aminoethyl)amine (0.585 g, 4 mmol). The mix solution was stirred for 24 h to form yellowish precipitate, which was isolated, washed with acetonitrile, and dried in vacuo. Weigh and dissolve the obtained yellowish solid (0.989 g, 1 mmol) in dichloromethane/methanol (DCM/MeOH, 50 mL/20 mL) in an ice bath, and then add sodium borohydride (0.38 g, 10 mmol) over the course of half an hour to reduce the imine cage. After stirring for a further six hours at room temperature, water was added, and the residue was extracted with DCM and washed with aqueous sodium carbonate (5%). The organic phase was combined and evaporated, which was added with 30 mL MeOH and 10 mL aqueous NaOH (1 M). After refluxing overnight, the obtain solution was first concentrated under reduced pressure, and then acidified with an excess amount of a 37% conc. HCl solution slowly to obtain a white precipitate, which was filtrated, and dried in vacuo to give a white powder (94%). $^1$H NMR (d-DMSO, 400 MHz): δ = 9.57 (s, 12H), 8.10 (s, 6H), 8.01 (d, $J$ = 8.5 Hz, 6H), 7.96 (d, $J$ = 8.5 Hz, 6H), 7.69 (s, 3H), 4.20 (s, 12H), 3.18 (s, 12H), 2.89 (s, 12H) ppm. $^{13}$C NMR (d-DMSO, 100 MHz): δ = 167.05, 143.09, 139.09, 132.72, 131.58, 130.09, 129.97, 129.21, 126.93, 49.80, 44.12 ppm. MALDI-TOF-HRMS (ESI): m/z 959.5178 [M + H]$^+$; found: 959.5156.

## Synthesis of protonated Cage-2

A mixture of the precursor aldehyde (1.260 g, 6 mmol) and tris(2-aminoethyl)amine (0.585 g, 4 mmol) in DCM/MeOH (50 mL/20 mL) was stirred at room temperature for 24 h. Sodium borohydride (0.38 g, 10 mmol) was then added over the course of half an hour. After the system had been stirred for a further six hours at room temperature, water was added and the residue was extracted with DCM and washed with aqueous sodium carbonate (5%). The organic solution was dried over sodium sulfate, evaporated, and dried in vacuo. The solution of precursor amine cage was acidified with 37% conc. HCl solution until a pH = 0 was reached. The resulting precipitate was isolated by filtration and dried in vacuo to give a white solid (91%). $^1$H NMR (D$_2$O, 400 MHz): δ = 7.95 (d, $J$ = 14.7 Hz, 2H), 7.84 (s, 2H), 7.79 (t, $J$ = 8.0 Hz, 1H), 7.57 (t, $J$ = 7.0 Hz, 2H), 7.36 (s, 1H), 4.26-4.30 (m, 4H), 4.19 (t, $J$ = 8.0 Hz, 4H), 3.43-3.37 (m, 4H), 2.86-2.91 (m, 4H) ppm. $^{13}$C NMR (D$_2$O, 100 MHz): δ = 134.26, 133.30, 132.02, 130.89, 129.75, 129.41, 128.63, 126.95, 51.50, 50.89, 46.22 ppm. MALDI-TOF-HRMS (ESI): m/z 828.5561 [M + H]$^+$; found:828.5598.

## Synthesis of protonated TA

A mix solution of (R,R)-1,2-diaminocyclohexane (1.14 g, 10 mmol), terephthalaldehyde (1.34 g, 10 mmol), methanol (100 mL) and triethylamine (3.5 mL, 25 mmol) was stirred overnight. The mixture was cooled in an ice bath, and added slowly with sodium borohydride (1.14 g, 30 mmol) over one hour. After stirring for three hours at room temperature, the solvents were removed under vacuum. The residue was then dissolved in DCM (50 ml) and washed with aqueous sodium carbonate, water and brine. The organic solution was combined, evaporated, and solubilized with ethanol. A solution of 10 mL of absolute EtOH with 2 mL of concentrated HCl was then added dropwise to form white precipitate, which was filtrated and dried to obtain the product as a white powder (95%). $^1$H NMR (D$_2$O, 400 MHz): δ = 7.39 (s, 4H), 4.43 (d, $J$ = 13.4 Hz, 2H), 4.07 (d, $J$ = 13.4 Hz, 2H), 3.46 (s, 2H), 2.37 (d, $J$ = 13.4 Hz, 2H), 1.85-1.75 (m, 2H), 1.59-1.47 (m, 2H), 1.33 (s, 2H) ppm. $^{13}$C NMR (D$_2$O, 100 MHz): δ 131.81, 129.98, 57.70, 48.27, 26.02, 22.15 ppm.

## Fabrication of humidity sensor

The Ag-Pd interdigital electrodes (IEs,) were deposited on the alumina ceramic substrate through a metal-jetting system, which consisted of a ceramic substrate (dimension: 13.4 mm × 7 mm × 0.635 mm) and fixed with 5 pairs of digits with 0.2 mm fingers and 0.2 mm gaps. To create the cage humidity sensors, cage aqueous solution were freshly prepared by dissolving the synthesized cage powder in water with a concentration of 10 mg/mL before use. The cage humidity sensors were fabricated through drop-casting followed by the activating method. 10 uL of cage aqueous solution (10 mg/mL) was drop-casted on the surface of the IEs, followed by drying at 70 °C for 1 min. The cage humidity sensor was activated by heating at 100 °C for 24 h under the vacuum condition. After that, a transparent cage-sensing film was successfully deposited on the surface of the IEs tightly.

## Assembly of touchless HMI devices

The 25 sensing units surface of the sensor array were deposited by Cage-1 solution, followed by drying at 70 °C for 1 minute. The cage humidity sensor array was activated by heating at 100 °C for 24 h under the vacuum condition. Subsequently, the sensor array is seamlessly integrated with the main circuit board, completing the sensory component of the system. Both sensory component and display component are powered by an electricity supply source, ensuring the continuous operation of both segments of the touchless HMI device.

## Humidity sensing test

We utilized various saturated salt solutions to generate well-controlled humid environment with certain RH levels in specialized chambers. Here, KNO$_3$, KCl, NaCl, CuCl$_2$, K$_2$CO$_3$, MgCl$_2$, CH$_3$COOK, and LiCl solutions produce 95%, 85%, 75%, 67%, 43%, 33%, 23%, and 11% RH conditions, respectively, which were monitored by a commercial hygrometer. The cage sensors' response to controlled different RH was evaluated by recording time-dependent resistance curves (R−t). The electrical measurements were carried out using a Keithley 2400 SourceMeter unit, with the sensor devices placed in hermetic chambers with the desired RH. All the gas sensing tests are monitored by keithley 2400 of resistance changes at room temperature (-25 °C), and each final data were obtained based on five times measurements. The dynamic response curves of resistance change versus time for cage sensors under various RH levels ranging from 11% to 95% were measured by switching the sensor from the low humidity chambers to the high humidity chambers one by one for an interval of 20 s. The humidity sensing curves of the humidity sensors response under different saturated salt solutions conditions and recovery under indoor relative humidity condition were measured by switching the sensor to the desired saturated salt solutions chambers when the test started, and then switched back to the indoor environment when the test ended. Humidity sensing curves with the RH changing from 11% to 95% were measured by switching the sensor to RH 11% chamber when the test started, and then switched back to the RH 95% chamber when the test ended.

## Proton conduction test

Impedance spectroscopy was employed for proton conduction measurements. T-shaped Teflon cells was assembled, with the pellets sandwiched between two platinum foil blocking electrodes. These cells was connected to an EC Labs Biologic VMP3 potentiostat using banana plug cables. Electrochemical impedance spectroscopy was conducted with a 2-probe setup, employing a sinusoidal perturbation of 10 mV across the frequency range of 1 MHz–10 mHz.

## Crystal structure and analysis

The crystals of Cage-2 suitable for single crystal diffraction were obtained from MeOH/water/DCM by slow evaporation at room temperature after several days. The crystals of Cage-1 and TA were

obtained by referring to their relevant single-crystal structures with the Cambridge Crystallographic Data Centre (CCDC) deposition numbers of 2179030 and 1868815, respectively.

Single crystal X-ray diffraction (SCXRD) data were recorded on a Bruker D8-Venture single crystal X-ray diffractometer equipped with a digital camera diffractometer using graphite-monochromated Mo- Kα radiation at 120 K. Data integration and reduction were performed using the SaintPlus 6.01 software. The absorption corrections and correction of other systematic errors were performed by the multi-scan method implemented in SADABS. Structures were solved using Direct Methods (SHELXS-97) and refined using the SHELXL- 2014[44] program package (full-matrix least squares on F$^2$) contained in OLEX2 and X-Seed[45,46]. In all cases the non-hydrogen atoms were refined anisotropically. The hydrogen atoms were fixed geometrically using riding atom model. The crystal data and refinement conditions for Cage-2 crystal is reported in Supplementary Tab 3. Further details could be obtained from the cif file, which could be obtained from the CCDC upon reference to CCDC number 2253396.

The molecular Hirshfeld maps with the fingerprint (2D) were calculated by using Crystal-Explorer 21.5. Hirshfeld surface analysis are assessed to obtain quantitative and qualitative insights into diverse type of intra or intermolecular interactions in a crystalline form of the researched compound. The Hirshfeld surface analysis is specified in the space occupied by a molecule in its crystal system based on the localized electron density distribution simulated around of spherical atom. The Hirshfeld surface via fingerprint plot is a powerful method to calculate the intermolecular interactions and show it with well-defined color codes.

Material Studio (version: 19.1) is used to analyze accessible surface, with a probe of 1 Å radius. Zeo$^{++}$ (version 0.3) was employed for void analysis, pore size distribution, and the interconnectivity of void space, where probe radii are 0.60 Å, 0.80 Å, 1.0 Å and 1.2 Å.

## Simulations

All simulations were performed with the CP2K code (version: 2022.2), which uses a mixed Gaussian/plane-wave basis set[47]. Double polarization quality Gaussian basis sets[48] and plane-wave cutoff of 400 Ry for the auxiliary grid were used. The density functional theory (DFT) calculations including geometry relaxation, single point energies and ab initio molecular dynamics (AIMD) simulations were conducted using the Perdew–Burke–Ernzerhof (PBE) exchange-correlation functional[49], with Grimme's D3 van der Waals correction (PBE + D3)[50]. AIMD simulations within the Born-Oppenheimer approximation were performed in the canonical (NVT) ensemble. A timestep of 1 fs was used for the integration of the equation of motion, and the simulations were run for 50 ps (50,000 AIMD steps following equilibration run that has a strong thermostat coupling). The temperature of the AIMD simulation was 500 K, which was controlled by the canonical sampling through velocity rescaling thermostat[51] using a time constant of 50 fs. The initial structures of Cage-1, Cage-2 and TA were taken from their crystal structures.

## Data availability

Crystallographic data in this study have been deposited free of charge in the Cambridge Crystallographic Data Centre under accession code CCDC No. 2179030 (Cage-1) [https://doi.org/10.5517/ccdc.csd.cc2c4g99], 2253396 (Cage-2), and 1868815 (TA) [https://doi.org/10.5517/ccdc.csd.cc20qncs]. All other data are available in the manuscript and Supplementary Information. Additional data are available from the corresponding author upon request. Source data are provided with this paper.

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

## Acknowledgements

N.M.K. acknowledge support from the King Abdullah University of Science and Technology (KAUST; grant OSR-2019-CRG8-4032).

## Author contributions

J.W. and W.L. contributed equally to this work. J.W. and W.L. conceived and designed the research project. W.L. designed and synthesized the organic cages. J.W. and W.L. characterized the materials. L.O.A. and W.L. natured and resolved the crystals. J.W., Z.C., and V.O.N. prepared humidity sensors and tested the sensing performance. J.W. designed and demonstrated the touchless HMI devices. W.L. performed calculations and simulations to reveal the mechanism. J.W. and W.L. analyzed the data and co-wrote the paper with the supervision of N.M.K. All authors discussed the results and commented on the manuscript.

## Competing interests

The authors declare no competing interests.
