## [Peer Review File · Nature Communications]

REVIEWER COMMENTS

Reviewer #1 (Remarks to the Author):

The authors introduce a novel application of molecular organic cage materials, an interesting addition to this field. While the study primarily centers around applied physics and engineering, the chemical characterization is meticulously executed, thoroughly analyzed, and adeptly presented. Notably, the Cage-1 molecule exhibits rapid responses and recovery in the face of humidity fluctuations, hinting at its promising practical applications. The authors have impressively manufactured sensing devices and illustrated a real-world HMI application, accompanied by comprehensive testing and in-depth analysis. Furthermore, the inclusion of control reactions across a range of relative humidity environments increased the robustness of this research.

In conclusion, this study underscores the successful integration of molecular organic cage materials into HMI devices. Beyond the demonstration of the HMI device's effectiveness, the authors have effectively conveyed their chemical and physical research. This manuscript is therefore publishable in NCOMM, providing the following comments can be addressed properly:

1. Despite the wealth of experiments presented, the manuscript's current version lacks a clear presentation of the mechanism underlying humidity sensing. Currently, it appears that structural differences lead to varying crystal packing models, ultimately impacting the critical factor of rapid humidity response, D_{water} . However, there is a notable discrepancy: Cage-1's resistance increases at RH 11% are significantly quicker than those of Cage-2 and TA, while TA's resistance decreases at RH 95% appear faster than Cage-2's. These results seem inconsistent with the trends observed in D_{water} . It is imperative that the authors provide a more comprehensive clarification, potentially backed by additional experimental results.
2. The chemical structures of Cage-2 and TA are relegated to the supplementary information, which impedes the clarity of the reading experience. Understanding the structural distinctions among Cage-1, Cage-2, and TA is pivotal for grasping their differing sensing capabilities. It is recommended that the authors incorporate the chemical structures into the main text for enhanced comprehension.
3. The initial paragraph of the results and discussion section introduces the concept of moisture-sensitive groups altering the proton conductivity of the cage material, resulting in electrical signal changes. However, this promising notion is not further explored in subsequent paragraphs, leading to confusion. If proton conductivity is indeed a pivotal aspect of humidity sensing, the authors should provide its characterization and propose a plausible mechanism for proton conduction in the three molecules studied. Alternatively, if proton conductivity is not a critical factor, it should be explicitly clarified in the manuscript.

4. The manuscript highlights the solution processability of molecular cage materials as a key advantage for humidity sensing. Nevertheless, the device presented is fabricated using a suspension of material, seemingly bypassing the touted solution processability. The authors may wish to address the solubility of Cage-1 and the potential for fabricating a humidity sensing device using suspensions of MOF/COF materials.

5. PXRD characterization is exclusively applied to Cage-1. It would greatly enhance the discussion comparing the three materials if the authors could provide PXRD results for Cage-2 and TA in the supplementary information to confirm their crystallinity.

6. The current mobile phone touchscreen sampling frequency exceeds 60 Hz, which is one to two magnitudes faster than the response rate demonstrated by the device in this work. It would be beneficial to include a table presenting benchmark data for response rates of various materials used in humidity sensing or touchless devices. Additionally, providing a conceptual outlook on how performance might be enhanced in future work would be valuable.

Reviewer #2 (Remarks to the Author):

In this paper, the author prepared a humidity sensor using a carboxyl and protonated amine functionalized organic cage. It exhibited excellent performance with the ultrafast response/recovery speed and excellent sensing stability, due to the strong interactions between –COOH groups in the organic cage and water confirmed by a large number of control experiments. Moreover, the authors fabricated a touchless fingertip humidity sensor with excellent performance, which can be used as a contactless password manager. This work seems carefully conducted and the manuscript is well organized. I suggest the publication of the manuscript after minor revision.

1. The author claims that the strong interactions between –COOH and water is the crucial for the excellent humidity sensing performance of cage-1, which makes it easy to understand that this humidity sensor has a fast response speed. However, the strong supramolecular interaction between cage-1 and water is difficult to disturb without external force. Why can cage-1 show ultra-fast recovery speed in a high-humidity environment? Could the author explain this?

2. Although the author proved that the Cage-1 has high stability, it is more concerned about the stability of the device in practical applications, and the author did not explain this. Will Cage-1 fall off after being made into a sensing film? How many times can the prepared sensor be recycled? Please add explanations.

3. The resistance values of the cage-1 humidity sensor when the humidity is 11%, 23%, and 33% are inconsistent in Fig. 2d and Fig. S11. Why?

4. When the Cage-1 sensor was continuously placed in the humidity range of 11%-95%, its resistance value (Fig. 2a) was significantly different from that when placed alone in different humidity environments (Fig. S11). Does this mean the sensor is unstable? Please explain this phenomenon.

5. The chemical structure of TA is incorrect in Figure S1, please correct it.

Reviewer #3 (Remarks to the Author):

The paper reports humidity sensors utilizing porous cages for touchless human-machine interaction. In this regard, touchless interaction based on humidity sensors does not exhibit obvious advantages over well-established magnetic-field, capacitive and ultrasonic touchless sensors in terms of response speed, interaction distance, and reliability. In particular, the sensitivity of humidity sensors may seriously deteriorate in high humidity environments. The main claim of the authors regarding the humidity sensor's performance is its fast response speed (1s/3s); however, this is considerably slower compared to many previously reported humidity sensors (e.g., ~30 ms, ACS Nano 2013, 7, 11166). Furthermore, the materials used in this study are not new, and the comparative studies presented in the paper lacks significance, the results are well expected. Given the well-established understanding about the influence of pore structure on humidity sensitivity, a more meaningful contribution would involve the development of new and novel pore structures capable of achieving record-breaking performance. Therefore, in my view, the novelty and significance of this paper fall short, not meeting the standards set by Nature Communications.

Other technical issues are listed below for authors to consider before publication.

1/ To make the research background more transparent, the authors may introduce all kinds of touchless sensors developed previously.

2/ The authors are encouraged to tabulate the performance of previously reported humidity sensors and put the table in SI.

3/ The thickness of the sensing layer was not clear, and the dimension of the sensors were missing.

4/ Can XRD analysis of the cages be conducted after the humidity sensing experiments to ascertain any structural changes?

5/ Figure S13 is blurry.

6/ Page 8, "To verify the accurate monitoring capacity of the sensor, a water surface humidity detector was developed based on the cage humidity sensor (Fig. 2c)." The experiments about the sensor's responses at different heights from the liquid level could not demonstrate the accuracy of the humidity sensor. Only Figure 6c which juxtaposes the test values with the actual values can illustrate the accuracy of the prepared sensor.

7/ The interaction distance is a crucial figure-of-merit. In Figure 5, it is observed that the fingers had to be in very close proximity to the sensor for a response to occur. However, the article does not specify the height at which the finger is positioned to elicit a response from the sensor. Calibration data correlating finger height with humidity response values should be included.

8/ Figure 5 demonstrates the non-contact sensing capability. However, in practical applications, the response occurs only when the finger is in close proximity to the sensing film. Once contact is made, there is a risk of erasing the sensing film, leading to sensor failure. We recommend optimizing the packaging structure of the humidity sensor to address this issue.

9/ Figure 5(a), the error bands are missing on the curves. In Figures (b), (c), and (d), the gray curve's error band appears to be substantial, why? The perceived large error band may indicate the low stability of the device. Consequently, it is essential to know whether the authors conducted exploratory experiments to identify the factors contributing to the large error band.

10/ Figure 6, can the authors perform the demonstrate in high humidity environment (over 90% HR)?

11/ In Figure S11, as the humidity transitions from indoor 60% RH to 95%, it is noticeable that the response time of the Cage-1 sensor exceeds 10 seconds. Does the specified response/recovery time (1s/3s) in the article exclusively apply to humidity changes from 11% to 95%? If so, it raises a concern about whether it (1s/3s) accurately represents the response time of the Cage-1 sensor.

12/ Figure S11d, why was the saturated resistance lower than that in Figures S11a,b,c?

Reply to Comments

Reviewer #1 (Remarks to the Author):

In conclusion, this study underscores the successful integration of molecular organic cage materials into HMI devices. Beyond the demonstration of the HMI device's effectiveness, the authors have effectively conveyed their chemical and physical research. This manuscript is therefore publishable in NCOMM, providing the following comments can be addressed properly:

Response

We thank the reviewer for the thorough evaluation of our manuscript. We deeply appreciate the positive feedback and constructive comments.

1. Despite the wealth of experiments presented, the manuscript's current version lacks a clear presentation of the mechanism underlying humidity sensing. Currently, it appears that structural differences lead to varying crystal packing models, ultimately impacting the critical factor of rapid humidity response, D_{water} . However, there is a notable discrepancy: Cage-1's resistance increases at RH 11% are significantly quicker than those of Cage-2 and TA, while TA's resistance decreases at RH 95% appear faster than Cage-2's. These results seem inconsistent with the trends observed in D_{water} . It is imperative that the authors provide a more comprehensive clarification, potentially backed by additional experimental results.

Response

We agree with the reviewer's comment and so we conducted proton conductivity measurements of Cage-1 and presented a clearer presentation of the mechanism underlying humidity sensing in the revised manuscript (revised Fig.2b and Fig.5). Moreover, as mentioned in the manuscript, multiple binding sites and 3D water channels of Cage-1 structure (Figure 4) endow the Cage-1 system with better water transport ability, which is the most important factor for the excellent performance of Cage-1. In Figure 5, we tried to explore the water transport behavior based on molecular dynamic simulations. The diffusion coefficients of water molecules (D_{water}) in Cage-1 ($D_{\text{water}} = 1.14 \times 10^{-3} \text{ cm}^2 \text{ s}^{-1}$) is nearly threefold of Cage-2 ($D_{\text{water}} = 4.05 \times 10^{-4} \text{ cm}^2 \text{ s}^{-1}$), and two orders of magnitude higher than that of macrocycle TA ($D_{\text{water}} = 4.52 \times 10^{-5} \text{ cm}^2 \text{ s}^{-1}$, Fig. 5b-d). The faster diffusion of water endow the system with quicker and better water recognition ability, namely easier to absorb/desorb the water molecules. In this regard, this result agrees well with Cage-1's resistance increase at RH 11% , which is significantly quicker than those of Cage-2 and TA. As for the comparison of TA and Cage-2, the scientific validity of comparing the performance of the two materials has been questioned. This is because these two materials exhibit significant data decay during cycling, indicating poor cyclic stability, and meaning that the rate of response and recovery is consistently changing significantly with cycling.

Therefore, it is difficult to make a direct performance comparison between these two materials. In the article, we specifically utilize the data from the first response cycle to show the superiority of Cage-1 compared to other structurally-related materials.

2. The chemical structures of Cage-2 and TA are relegated to the supplementary information, which impedes the clarity of the reading experience. Understanding the structural distinctions among Cage-1, Cage-2, and TA is pivotal for grasping their differing sensing capabilities. It is recommended that the authors incorporate the chemical structures into the main text for enhanced comprehension.

Response

We have incorporated the chemical structures of Cage-2 and TA into the main text, per the reviewer's suggestion.

3. The initial paragraph of the results and discussion section introduces the concept of moisture-sensitive groups altering the proton conductivity of the cage material, resulting in electrical signal changes. However, this promising notion is not further explored in subsequent paragraphs, leading to confusion. If proton conductivity is indeed a pivotal aspect of humidity sensing, the authors should provide its characterization and propose a plausible mechanism for proton conduction in the three molecules studied. Alternatively, if proton conductivity is not a critical factor, it should be explicitly clarified in the manuscript.

Response

We appreciate your insightful suggestion. Proton conductivity is indeed a pivotal aspect of humidity sensing. In accordance with your suggestion, we have performed a comprehensive test of the proton conductivity of the sensing material and have described it in detail in the revised manuscript as highlighted (page 8, revised Figure 2, Table S1). Moreover, to propose a plausible mechanism for proton conduction in the three molecules studied, we measured their proton conductivity by using alternating current electrochemical impedance spectroscopy (EIS) at different temperatures (303–343 K) under RH of 85% (Fig.5a-c, Supporting Information Table S4) and determined the different conduction mechanisms of the three materials by calculating the activation energy (Fig. 5d).

4. The manuscript highlights the solution processability of molecular cage materials as a key advantage for humidity sensing. Nevertheless, the device presented is fabricated using a suspension of material, seemingly bypassing the touted solution processability. The authors may wish to address the solubility of Cage-1 and the potential for fabricating a humidity sensing device using suspensions of MOF/COF materials.

Response

Thank you for your valuable suggestion. Unlike many other COF and MOF materials, the cage material in our study exhibits exceptional water solubility (Figure S8). This unique characteristic significantly simplifies the preparation of cage sensors, making the process notably straightforward. We realize the confusion caused by using the term "suspension". Per

the reviewer's comment, the term "suspension" has been replaced by "aqueous solution" and we have modified Fig. S8 accordingly.

5. PXRD characterization is exclusively applied to Cage-1. It would greatly enhance the discussion comparing the three materials if the authors could provide PXRD results for Cage-2 and TA in the supplementary information to confirm their crystallinity.

Response

We have included the PXRD results for Cage-2 and TA in the supplementary information (Fig.S7), per the reviewer's suggestion.

6. The current mobile phone touchscreen sampling frequency exceeds 60 Hz, which is one to two magnitudes faster than the response rate demonstrated by the device in this work. It would be beneficial to include a table presenting benchmark data for response rates of various materials used in humidity sensing or touchless devices. Additionally, providing a conceptual outlook on how performance might be enhanced in future work would be valuable.

Response

We thank the reviewer for this important comment. In our revised manuscript, we have incorporated a table in Supporting Information (Table S2) presenting data for response rates and cyclic performance of various materials used in humidity sensing, as shown below.

Table S2 Humidity sensing performances comparison of different humidity sensors.

Material	Type	Humidity range (% RH)	Response time(s)	Recovery time(s)	Cyclic performance (cycles)
Er doped ZnO nanoparticles	Impedance	11-95	32	39	3
Cu doped ZnO thin film	Resistive	15-95	32	47	-
KCl-doped TiO2 nanofibre	impedance	11-95	3	3	-
Mg ²⁺ /Na ⁺ - doped TiO2 nanofiber	impedance	11-95	2	1	10
SnO2 dodecahedral nanocrystals	impedance	11-95	4	13	3
SnO2-WS2 nano-composite	Resistive	11-95	-	50	5
LiCl-Pebax 2533	impedance	11-95	30	80	-
SBA-15-PSS	impedance	11-95	5	106	5
keratin/graphene oxide	impedance	16-92	41	62	-
Graphene Oxide	impedance	15-95	10.5	41	several
PEVIm-Br	Resistive	11-98	6	31	4
Nanofiber of SPEEK	impedance	11-98	1	25	5
PAM/cassava gum- polyol -LiBr	impedance	11-98	275.6	227.0	4
PVA/CNF	impedance	11-98	380	140	3
Graphene Oxide	Resistive	12-97	50	421	5
PANI/NFC/PVA	Capacitive	30-100	47	58	4
PEDOT: rGO-PEI/Au	Resistive	11-98	20	-	<100

Graphene	Resistive	1-96	0.6	0.4	3
Pyranine-rGO	impedance	11-95	2	6	100
RGO-BiVO4 heterojunction	impedance	11-95	3.6	18	30
Li/K-codoped 3DOM WO3	impedance	11-95	15	10	5
Ag/SnO2	impedance	11-95	4	6.5	2
Graphene Oxide	Capacitive	10-90	15.8	-	6
[P(VDF-TrFE) nanocone arrays	Capacitive	50-90	3.7	3.4	3
Silk fibroin	Resistive	43-95	73.1	11.3	5
MoO3	Resistive	0-100	0.5	2	5
Pd/HNb3O8	Resistive	30-99.9	0.2	3	50
VS2	Resistive	0-100	30-40	12-50	4
WS2	Resistive	11-97	12	13	4
Pt-nRGO fiber	Resistive	6.1-66.4	0.064	0.508	30
TiO2 nanowire	Voltage	20-90	4.5	2.8	5
MWCNTs/PAA	Resistive	50-90	680	380	3
MWCNTs/ PVP	Resistive	11-94	15	1.8	3
MnO2-coated CNT yarn	Resistive	65-90	20	30	5
N-doped carbon spheres	Impedance	9-97	19	178	3
PDDA/rGO	Resistive	11-97	108-147	94-133	15
Au/GO/silica	Impedance	20-90	119	125	-
15 nm Graphene Oxide	Impedance	10-90	30ms	30ms	5
Cage (This work)	Resistive	11-95	1	3	800

Our humidity sensor demonstrates exceptional sensing speed and remarkable cyclic stability, essentially meeting the criteria for touchless HMI applications. It is important to note that the sampling frequency of the device is intricately linked to both the response rate of the sensing material and the configuration of the associated hardware equipment. Achieving optimal functionality necessitates a meticulous match between these components. Ultimately, based on the reviewer's suggestion, we included a conceptual outlook on how the performance might be enhanced in future work (highlighted in the revised manuscript).

Reviewer #2 (Remarks to the Author):

In this paper, the author prepared a humidity sensor using a carboxyl and protonated amine functionalized organic cage. It exhibited excellent performance with the ultrafast response/recovery speed and excellent sensing stability, due to the strong interactions between –COOH groups in the organic cage and water confirmed by a large number of control experiments. Moreover, the authors fabricated a touchless fingertip humidity sensor with excellent performance, which can be used as a contactless password manager. This work seems carefully conducted and the manuscript is well organized. I suggest the publication of the manuscript after minor revision.

Response

Thank you for your positive feedback and thoughtful evaluation of our manuscript.

1. The author claims that the strong interactions between –COOH and water is the crucial for the excellent humidity sensing performance of cage-1, which makes it easy to understand that this humidity sensor has a fast response speed. However, the strong supramolecular interaction between cage-1 and water is difficult to disturb without external force. Why can cage-1 show ultra-fast recovery speed in a high-humidity environment? Could the author explain this?

Response

The introduction of three carboxylic acid groups endows Cage-1 with some distinct differences. 3D structure with more abundant water channels, and the higher capacity of water molecules binding sites.

- These –COOH groups interact with chloride anions to expand the void space in the crystal, which thereafter brings about the 3D interconnected water channels that is easier for water molecules to transport/diffuse (Fig 4a).
- Although there is no obvious interactions between –COOH and water molecules regarding the Cage-1 crystal structure, the –COOH could work together with $-NH_2^+$ groups to interact with water molecules in the real sensor systems because they are both good hydrogen bonding functional groups.
- Quantitative analysis based on calculated Hirshfeld surfaces also revealed that hydrogen bonding $O\cdots H$ play an important role in Cage-1 (20.9%), which is more than twice of TA (8.5%) and nearly four-fold of Cage-2 (5.5%). This result indicates much more water affinity to Cage-1, which could be a contribution of –COOH to both structural improvement and hydrogen binding sites increase. Moreover, the additional proton conductivity and activation energy calculation (Fig. 5) further revealed the Grotthuss mechanism in the Cage 1, indicating better proton conductivity due to more binding sites from –COOH groups Thus, it is more accurate to conclude that stronger supramolecular interactions between water molecules and Cage-1 host because of the introduction of –COOH to realize better structure for water diffusion and more water binding sites.

2. Although the author proved that the Cage-1 has high stability, it is more concerned about the stability of the device in practical applications, and the author did not explain this. Will Cage-1 fall off after being made into a sensing film? How many times can the prepared sensor be recycled? Please add explanations.

Response

Thank you for raising this important consideration regarding the stability of the device in practical applications.

1) In response to your suggestions, we subjected the device to cyclic stability and time stability tests. After hundreds of demonstrations, the device displayed excellent repeatability. Impressively, even after half a year, the device continued to exhibit remarkable repeatability. (Highlight in Page 19)

2) To assess the robust durability of the sensor, an abrasion test was initiated, subjecting the sensing film to rubbing by a finger dozens of times. Subsequently, no detachment of the sensing film was observed as shown video 2 in the Supporting Information. The sensing film exhibited commendable interfacial adhesion to the sensor substrate after drop-casting and drying, confirming its robust durability. Subsequent finger humidity sensing test were also performed and the sensing signal was not attenuated by contact friction (Fig. S26), which proves the excellent stability of the Cage-1 sensor. (Highlight in Page 18)

These results have been added in the revised manuscript to provide a comprehensive understanding of the stability performance of our sensing devices.

3. The resistance values of the cage-1 humidity sensor when the humidity is 11%, 23%, and 33% are inconsistent in Fig. 2d and Fig. S11. Why?

Response

The inconsistency in resistance values can be attributed to the following two factors:

1) The humidity hysteresis effect: Humidity hysteresis refers to the phenomenon where the moisture content in a material or the response of a humidity sensor does not follow the same path during the process of adsorption and desorption. In other words, the humidity level at which a material absorbs moisture might be different from the humidity level at which it releases moisture. Fig. 2a illustrates the continuous response of the material to different humidity levels, starting at RH 11%. The resistance shows a stepwise decrease with increasing humidity (23%, 33%, or even higher), indicative of a typical moisture adsorption process. In contrast, Fig. S11 represents isolated sensor tests focused on specific humidity levels. The purpose is to compare the response rate at various humidity levels. These tests initiate from an environmental humidity level (~60% RH) and transition to target other values (e.g., 11%, 23%, and 33%). During this transition, the initial humidity is higher than the target humidity, representing a typical moisture desorption process. Therefore, even at the same humidity values in Fig. 2a and Fig. S11, the unavoidable humidity hysteresis effect in both adsorption and desorption processes leads to differences in resistance values.

2) The different test conditions and kinetics involved in these two scenarios: In Fig. 2a, we conducted a continuous transfer from humidity levels of 11% to 23% and 33%. The device

was exposed to varying levels of humidity for about 20 seconds. And it should be noted that during the dynamic transfer process it is inevitable to pass briefly through ambient humidity state (relative humidity value of 60%). So, the sensor experiences a continuous and dynamic change in humidity levels, leading to complex and overlapping responses. However, Fig. S11 illustrates an alternating test where the sensor transitions from an ambient humidity state to different humidity levels. The device was exposed to varying levels of humidity (humidity levels of 11% to 23% and 33%) until the resistance reach to the highest value. The kinetics of the response reflected in the two different test methods are completely different. Therefore we can understand that the response and recovery time will be different for these two different test methods.

4. When the Cage-1 sensor was continuously placed in the humidity range of 11%-95%, its resistance value (Fig. 2a) was significantly different from that when placed alone in different humidity environments (Fig. S11). Does this mean the sensor is unstable? Please explain this phenomenon.

Response

As we mentioned in response to the above question, the variation in resistance values between Fig. 2a and Fig. S11 is attributed to the humidity hysteresis effect and different test conditions and kinetics. However, this does not impact the sensing stability of our material, as confirmed by continuous testing over 800 cycles without performance degradation, as shown in Fig. S14, a rarity in the published literature.

5. The chemical structure of TA is incorrect in Figure S1, please correct it.

Response

In the revised version, we have corrected the chemical structure of TA and included the updated structure in Fig. 3c.

Reviewer #3 (Remarks to the Author):

The paper reports humidity sensors utilizing porous cages for touchless human-machine interaction. In this regard, touchless interaction based on humidity sensors does not exhibit obvious advantages over well-established magnetic-field, capacitive and ultrasonic touchless sensors in terms of response speed, interaction distance, and reliability. In particular, the sensitivity of humidity sensors may seriously deteriorate in high humidity environments. The main claim of the authors regarding the humidity sensor's performance is its fast response speed (1s/3s); however, this is considerably slower compared to many previously reported humidity sensors (e.g., ~30 ms, ACS Nano 2013, 7, 11166). Furthermore, the materials used in this study are not new, and the comparative studies presented in the paper lacks significance, the results are well expected. Given the well-established understanding about the influence of pore structure on humidity sensitivity, a more meaningful contribution would involve the development of new and novel pore structures capable of achieving record-breaking performance. Therefore, in my view, the novelty and significance of this paper fall short, not meeting the standards set by Nature Communications.

Response

We appreciate the reviewer's comments and respect the expressed opinion.

While it is true that magnetic-field, capacitive, and ultrasonic touchless sensors are well-established technologies, our study aims to explore an alternative and complementary avenue for touchless sensing, which goes way beyond the detection scope of other magnetic-field, capacitive and ultrasonic touchless sensors. The advantages of humidity sensors in touchless HMI stem from their ability to capture and respond to a unique and natural aspect of human interaction—skin moisture. Human skin naturally releases moisture, and humidity sensors can capture variations in this moisture. This allows for a more natural and intuitive form of touchless interaction, as it leverages a biological parameter. Moreover, we have evaluated the sensor's resistance to high humidity levels. Specifically, we exposed the sensor to 95% RH for 36 hours. The results indicated that the sensor exhibited robust performance under these conditions, with no discernible degradation observed. As for response speed and recovery, we have incorporated a table in supporting information (Table S2) presenting data for response rates of various materials used in humidity sensing. It proves that Cage-1 demonstrates exceptional sensing speed and remarkable cyclic stability, essentially meeting the criteria for touchless HMI applications. While it's accurate that certain humidity sensors have achieved faster response times, it's crucial to consider the specific application context and the trade-offs involved. We also have to consider the material's cyclic stabilization. The novelty of our work is not just heavy on the fabrication of the humidity sensor, but more importantly lies in the successful application of molecular building blocks for HMI, in which reported response speed meets the requirements for various real-world scenarios. We have revised our manuscript to better showcase the novelty (Highlighted in Page 3).

Other technical issues are listed below for authors to consider before publication.

1/ To make the research background more transparent, the authors may introduce all kinds of touchless sensors developed previously.

Response

Thank you for your valuable suggestion. In the revised manuscript, we expanded the introduction to include a thorough review of different types of touchless sensors that have been developed in the past (Highlighted in the revised introduction).

2/ The authors are encouraged to tabulate the performance of previously reported humidity sensors and put the table in SI.

Response

We have created a table summarizing the performance metrics of previously reported humidity sensors in the Supplementary Information Table S2, per the reviewer's suggestion.

3/ The thickness of the sensing layer was not clear, and the dimension of the sensors were missing.

Response

We employed additional methods to assess the thickness of the sensing film. Initially, a spiral micrometer was utilized, revealing a thickness of approximately 6 μm . Subsequently, we conducted a more precise measurement of the sensing film thickness using SEM, yielding an thickness of about 6.17 μm (Fig.S11). This dual-method approach substantiates the accuracy of our thickness measurements. Also, we can see from the SEM photos that the sensing film is very uniform.

4/ Can XRD analysis of the cages be conducted after the humidity sensing experiments to ascertain any structural changes?

Response

We thank the reviewer for this insightful suggestion. We have conducted the PXRD for the cage before and after the humidity sensing experiments in Fig S17. The results indicate no significant change in the structure of the Cage-1 before and after the humidity sensing experiment (Fig. S17). This further validates the structural stability and ultra-high cyclic stability of the Cage-1.

5/ Figure S13 is blurry.

Response

This was revised accordingly. Our apologies for the low original quality.

6/ Page 8, “To verify the accurate monitoring capacity of the sensor, a water surface humidity detector was developed based on the cage humidity sensor (Fig. 2c).” The experiments about the sensor’s responses at different heights from the liquid level could not demonstrate the accuracy of the humidity sensor. Only Figure 6c which juxtaposes the test values with the actual values can illustrate the accuracy of the prepared sensor.

Response

The intention behind humidity detection on the water surface was to highlight the varying responsiveness of humidity sensors under different humidity conditions. It is evident from the data that as the sensor height increases above the water surface, both humidity values and the sensor's response values decrease. These data indeed do not provide a basis for accurately judging the sensor's accuracy but rather demonstrate the sensor's responsiveness under different humidity conditions. In response to this clarification, we have made appropriate adjustments in the revised manuscript to rectify any inaccuracies in the original statement.

7/ The interaction distance is a crucial figure-of-merit. In Figure 5, it is observed that the fingers had to be in very close proximity to the sensor for a response to occur. However, the article does not specify the height at which the finger is positioned to elicit a response from the sensor. Calibration data correlating finger height with humidity response values should be included.

Response

We appreciate this valuable suggestion. We have included this data in the revised manuscript (Fig.6a, highlighted page 17) to offer a more comprehensive understanding of the sensor's performance in relation to the distance of interaction.

8/ Figure 5 demonstrates the non-contact sensing capability. However, in practical applications, the response occurs only when the finger is in close proximity to the sensing film. Once contact is made, there is a risk of erasing the sensing film, leading to sensor failure. We recommend optimizing the packaging structure of the humidity sensor to address this issue.

Response

For all demonstrations, the distance between the finger and the sensor is controlled at 4-6 mm. In accordance with your suggestion, we firstly initiated an abrasion test on the sensor, subjecting the sensing film to vigorous rubbing by finger up to dozens of times. After that, we didn't notice any detachment of the sensing material (from Video 2 in Supporting Information). The material exhibited commendable interfacial adhesion to the sensor substrate after drop-casting and drying, affirming its robust durability. Subsequent finger

humidity sensing tests were also performed and the sensing signal was not attenuated by contact friction (Fig.S26). This proves that our sensor has very excellent stability. We have incorporated the data into the revised manuscript.

9/ Figure 5(a), the error bands are missing on the curves. In Figures (b), (c), and (d), the gray curve's error band appears to be substantial, why? The perceived large error band may indicate the low stability of the device. Consequently, it is essential to know whether the authors conducted exploratory experiments to identify the factors contributing to the large error band.

Response

The average mean square-displacements (MSDs) calculated in this dynamic simulation in Fig 5a, were mainly to explore and compare the different water diffusion ability in the three materials with different water channels. Namely, for better comparison, we just show the trend of MSDs in Fig 5a, so the detailed simulation results were shown separately in Fig 5b-d. The purpose of error band added here is to show the possible scope, help to show the MSD trend, and make the results more precise. In fact, the range of error band is still in the same magnitude to the average value. This is reasonable for this kind of ab initio molecular dynamics (AIMD) simulations for water molecules in 500 K, in which water molecules are in vigorous movement. For reference, similar error band were seen in Fig 5, ref. 33 (Yang, Z. et al. Supramolecular Proton Conductors Self-Assembled by Organic Cages. *JACS Au* 2, 819–826 (2022)).

10/ Figure 6, can the authors perform the demonstrate in high humidity environment (over 90% HR)?

Response

The real-life conditions where the fingertip sensitivity was tested in Fig. 6 is around 60%, which is ideal for this type of application. It's crucial to emphasize that the relative humidity values around the human finger exceed those of the test environment, creating a prerequisite for the effective operation of our caged finger humidity sensor. Therefore, testing the device at humidity exceeding 90% will not meet the prerequisites for the normal operation of the device. Fortunately, in typical living environments, relative humidity values are usually lower than the humidity around the fingers. Consequently, our devices are well-suited for seamless operation under normal living conditions.

11/ In Figure S11, as the humidity transitions from indoor 60% RH to 95%, it is noticeable that the response time of the Cage-1 sensor exceeds 10 seconds. Does the specified response/recovery time (1s/3s) in the article exclusively apply to humidity changes from 11% to 95%? If so, it raises a concern about whether it (1s/3s) accurately represents the response time of the Cage-1 sensor.

Response

We appreciate your careful examination of the data and your insightful question. Describing how quickly a sensing material reacts requires clear details about the test conditions, as the material's response can vary depending on the situation. Take Figure S11 as an example, where we show experimental results for how the material responds in different test humidity levels and recovers in regular ambient humidity conditions. The material's speed in responding is closely tied to the difference between the test humidity and the regular humidity around it. In short, the greater the difference, the faster the response. This is because a larger humidity gradient between the material and the surroundings causes the water molecules to move quickly between the material and the surroundings, creating a rapid diffusion. Looking at the difference in response time under two test conditions – changing humidity from 60% to 95% and changing humidity from 11% to 95% – highlights how specific testing details impact the material's performance. So, when we mention a response/recovery time of (1s/3s), it's important to note that this specifically applies to situations where the humidity changes from 11% to 95%. We've made this clearer in our revised explanation as highlighted in the revised manuscript.

12/ Figure S11d, why was the saturated resistance lower than that in Figures S11a,b,c?

Response

For humidity levels of 11%, 23%, and 33%, although the final resistance can reach the saturated resistance, the time taken to reach this value significantly differs. However, at 43% humidity, the driving force is insufficient to cause the material to reach the resistance of $200\text{M}\Omega$ within a specific timeframe. That is why the saturated resistance was lower than that in Figures S11a,b,c.

REVIEWERS' COMMENTS

Reviewer #1 (Remarks to the Author):

The authors have diligently addressed reviewer comments, incorporating additional data, explanations, and corrections to enhance the clarity of the article. The inclusion of additional information in the revised introduction has significantly improved the overall presentation. The revised paper meets well the scientific quality of Nat. Commun., which is therefore recommended for acceptance.

Reviewer #2 (Remarks to the Author):

Thanks to the authors for trying to carefully answer the questions and the extra effort to improve the manuscript, and I would be happy to support its acceptance.

Reviewer #3 (Remarks to the Author):

This reviewer has no further comments and respect the decision of editor.